# Relationships between Local Green Space and Human Mobility Patterns during COVID-19 for Maryland and California, USA

**Seulkee Heo \*, Chris C. Lim and Michelle L. Bell**

School of the Environment, Yale University, New Haven, CT 06511, USA; chris.lim@yale.edu (C.C.L.); michelle.bell@yale.edu (M.L.B.)

**\*** Correspondence: seulkee.heo@yale.edu

**Abstract:** Human mobility is a significant factor for disease transmission. Little is known about how the environment influences mobility during a pandemic. The aim of this study was to investigate an effect of green space on mobility reductions during the early stage of the COVID-19 pandemic in Maryland and California, USA. For 230 minor civil divisions (MCD) in Maryland and 341 census county divisions (CCD) in California, we obtained mobility data from Facebook Data for Good aggregating information of people using the Facebook app on their mobile phones with location history active. The users' movement between two locations was used to calculate the number of users that traveled into an MCD (or CCD) for each day in the daytime hours between 11 March and 26 April 2020. Each MCD's (CCD's) vegetation level was estimated as the average Enhanced Vegetation Index (EVI) level for 1 January through 31 March 2020. We calculated the number of state and local parks, food retail establishments, and hospitals for each MCD (CCD). Results showed that the daily percent changes in the number of travels declined during the study period. This mobility reduction was significantly lower in Maryland MCDs with state parks (*p*-value = 0.045), in California CCDs with local-scale parks (*p*-value = 0.048). EVI showed no association with mobility in both states. This finding has implications for the potential impacts of green space on mobility under an outbreak. Future studies are needed to explore these findings and to investigate changes in health effects of green space during a pandemic.

**Keywords:** mobility; vegetation; green space; sustainability; social media; disease prevention

## 1. Introduction

The COVID-19 outbreak occurred just before the 2020 Lunar New Year in China [1] and rapidly led to a global spread. The World Health Organization (WHO, Geneva, Switzerland) declared that the new coronavirus outbreak is an international public health concern on 30 January 2020 [2] and WHO officially announced COVID-19 as a pandemic on 11 March 2020 [3]. On 16 March 2020, the US federal government announced the '30 Day to Slow' guideline in response to the pandemic. While implementation of mitigation measures varied by US state and district, many of those declarations were announced in March [4]. For example, Maryland enacted a "Prohibiting Large Gatherings and Events and Closing Senior Centers" order on 12 March 2020; a statewide "stay-at-home" order went into effect on 30 March [5]. California declared a state of emergency in March and a stay-at-home order in 19 March. Under these orders, residents were permitted to go outside "for fresh air and exercise as long as they are maintaining a safe distance from others."

Social distancing, or physical distancing, has widespread consequences, affecting the economy and individuals' behaviors in various ways [6], including significant decreases in human mobility and

traffic volume around the period when various governments announced interventions (i.e., guidelines for social distancing, quarantine, and stay-at-home orders) [1,7,8]. While the definition of mobility varies among studies and disciplines, mobility analysis examines possible travel destinations and travel route based on local land use and demographics [9]. A better understanding of the mobility (e.g., destinations) of people can assist decision-making in prevention of disease transmission [10]. For this, studies showed that aggregated human mobility based on mobile phone data can assist studies assessing the spread of epidemics [11], economic consequence of the COVID-19 pandemic [6,12], and how the stay-at-home orders are effective to mitigate human mobility and thereby reduce the COVID-19 transmission [1,13,14]. Large mobility reduction was detected following the COVID-19 pandemic and specific government directives in the US and globally [15–17]. A study using mobility data from Wuhan and transmission of cases across China found that the positive relationship between human mobility and COVID-19 cases decreased after control measures [1]. A few other studies also suggested that sustained human mobility due to domestic and/or international air travel bans contributed to decreased transmission of COVID-19 cases at the early stages of the outbreak in European countries [18] and China [19]. Given the clear links between mobility and spread of the novel coronavirus [20], understanding mobility patterns is crucial to address COVID-19 outbreaks and develop policies to minimize transmission. While human mobility data have been utilized in some scientific works regarding visualizing mobility patterns [21] and economic effects of mobility changes [6], little is known about whether and how environmental factors influence human mobility under normal conditions and during the COVID-19 pandemic.

Alongside government directives for staying at home and social/physical distancing, health authorities including U.S. Centers for Disease Control and Prevention (CDC) and WHO emphasized the importance of regularly performing exercise to cope with the stress of quarantine, stay healthy, and maintain immunity [22]. Several studies argued that physical inability as a consequence of strict quarantine may be associated with increased risk of mental health outcomes as well as cardiovascular diseases, metabolic diseases, and cancer [23–26]. Thus, the COVID-19 pandemic along with other major environmental crises such as climate change shed a light on the need for better understanding of how to promote resilience or capacity of societies to deal with complex health crises [26].

Earlier work indicates that green space provides health benefits [25,27] and sustainability in cities [28]. Green space is defined as natural vegetation such as grass, bush, plants or trees and the built green structures such as parks and unstructured vegetated areas [29]. Potential pathways for the health benefits from green space include encouraging physical activities and providing direct interactions with nature [30]. Given the restrictions on the gathering of people particularly in indoor settings during COVID-19, understanding the use of green space contributes to our understanding of how green space relates to the ability of communities to cope with the stress from quarantine and pandemic, such as by playing a role as an alternative place for physical activity. A recent study in Oslo, Norway found that outdoor physical activity levels increased after the lockdown was implemented, and that the increases were highest in trails with greener and more remote areas [31]. A study conducted in the US found that the reduction in mobility to parks impacted by state-of-emergency declarations was smaller than the mobility reduction for other venues across the states [32]. Thus, green space could be an effective modifier on the effectiveness of COVID-19 mitigation measures, and such measures could indirectly impact the public health benefits of greenness.

As of April 2020, most US states have ordered nonessential businesses such as restaurant, bars, theaters, and gyms to close but the status of green spaces and open spaces such as parks have been far less consistent in many states. Green space may be one of the limited outdoor places where people seek to perform outdoor exercise during the COVID-19 crisis. The aim of this study was to investigate green space as a potential factor influencing mobility patterns during a pandemic. We hypothesized that the expected mobility decreases due to the social/physical distancing and associated policies will be lower in areas with higher local green space. Specifically, for a case study region of Maryland and California, we examined how the temporal trends in the number of people traveling among the study

regions (minor civil division (MCDs) for Maryland, census county division (CCD) for California) differ by local vegetation level. The results here provide information relevant for the design and effectiveness of sheltering policies designed to mitigate a pandemic.

## 2. Materials and Methods

We utilized de-identified and aggregated large-scale data developed by Facebook Data for Good platform to identify population mobility trends during the COVID-19 crisis [33]. These data are called Movement Data and they aggregate information from people using the Facebook app on their mobile phones with location history turned on to show movements between two points. The computation system of this data acquires the most common location of the Facebook app user using Bing tile map Level 13 (e.g., approximately 4.9 × 4.9 km) [34] within the first time window and the most common location in the second time window for each day. The centroids of the starting and ending Bing tiles are assigned to the person's movement vector. Then, these vectors are aggregated into the administration level 4 boundary (e.g., township), which is spatially equivalent to MCDs in Maryland and CCDs in California. MCDs and CCDs are administrative county subdivisions for which the Census Bureau establish and provide subcounty statistics [35]. CCDs are equivalent geographic entities to MCDs in US states where MCDs do not exist or have been unsatisfactory for comparing statistical data [35]. The data characterized mobility trend as the percent change in the observed number of users traveling into the administrative area (MCD or CCD) for the same time window and the same day of the week compared to the baseline period at MCD (CCD) level so the mobility data indicate movement of users across regions at this spatial level. The baseline number of users moving into an MCD (CCD) was calculated as the average number of users moving for the same daytime window and day of the week between 26 February 2020 to 10 March 2020. The percent change in the number of users between a given day and its baseline is calculated as follows [33]: percent change = $\left(c_t - \mu_{baseline,t}\right/\left(\mu_{baseline,t} + \epsilon\right)$, where $c$ is the number of users moving into a MCD (CCD) for day $t$, $\mu_{baseline,t}$ is the mean of the number of users traveling into the same MCD over the same time interval on the same day of the week of day $t$ during the baseline period, and $\epsilon$ is a small value, in this case 1. The distance traveled by users for each MCD (CCD) was calculated as the distance of the movement vectors linking the centroids of the starting and end Bing tiles of all users who traveled into that MCD (CCD). We used the daytime window (8 a.m.–4 p.m.) for our analysis.

The Movement Data of our study regions were linked to the geographic information systems (GIS) data of MCD in Maryland and CCD in California provided by the US Census Bureau. We analyzed the 230 administrative areas (MCDs) out of 290 total administrative areas in the state of Maryland, US, for which the Movement Data were available, in order to examine mobility trends for the period from 10 March to 24 April 2020. For California, we analyzed 341 CCDs out of 397 CCDs, for which the Movement Data were available. Percent change in the number of users moving was not observed for some MCDs when MCDs are smaller than a Bing tile so the user's location at MCD and CCD level cannot be specified. Data are not provided for county subdivisions where the number of observed users is smaller than 10 users to protect users' privacy. County subdivisions with no observation for users' moving between pairs of county subdivisions for 70% or more of the days in the study period were excluded in the analysis. As a result, 76 MCDs in Maryland and 241 CCDs were included in our main analysis.

Vegetation level to indicate green space was estimated by the Enhanced Vegetation Index (EVI) from the Moderate Resolution Imaging Spectroradiometer (MODIS) product MOD13Q1, which is a 16-day composite image at 250-meter resolution. The EVI is an advanced version of the Normalized Different Vegetation Index (NDVI). The NDVI is calculated as near-infrared radiation minus visible radiation divided by near-infrared radiation plus visible radiation (i.e., NDVI = (NIR − RED)/(NIR + RED)). The index ranges from −1 to +1 with higher values indicating denser vegetation and −1 indicating waterbody features (NASA, 2018). While the EVI's calculation is similar to NDVI, it corrects for some distortions in the reflected light caused by the particles in the air, ground cover below the

vegetation, and the saturating effects of areas with large amount of chlorophyll such as rainforests [36]. We calculated the average EVI for each study region (MCD, CCD) using the EVI pixel values within and surrounding the MCD and CCD boundary, for 1 January 2020 through 21 March 2020 to represent vegetation level in the study regions.

The MODIS Land Cover Type Product (MCD12Q1) [37] was used to estimate urbanicity of each study region. The number of 'Urban and Built-up Lands' pixels based on the University of Maryland legend and class definition was divided by the total number of pixels within the MCD or CCD boundary was calculated as the percent impervious area.

We obtained the park data in California from Esri Data & Maps [38]. These data include parks and forests at national, state, county, and local levels (e.g., city-scale parks, pocket parks, playgrounds, etc.) [38]. We obtained the Maryland State Parks data provided by the OpenStreetMap [39]. The OpenStreeMap is a global dataset including open user-generated street maps, geographical features, and built environment, which have been used for a wide range of studies [40]. This Maryland State Parks data, developed by the Baltimore County Government, is a GIS shape file and includes geographical polygons for several types of green space: state and national parks or forest, hiking trails, natural resource management areas for recreational activities (e.g., fishing, hunting, wild animal observation, etc.) and preservation of environmental resources, and wildlife management areas [41]. Data of parks at local levels in Maryland were obtained from Esri Data & Maps [38]. Hereafter, we use the term parks to refer to all these types of areas. The continuous variable for parks did not have a normal distribution. Thus, we considered a categorical variable of presence of parks within MCDs and CCDs (i.e., MCDs with state parks vs. MCDs without state parks within their boundary).

We obtained the GIS file of food retail establishments and hospitals from various sources. We obtained the data for food stores (2017–2018) and restaurants (2019) in Maryland from the Maryland Food System Map (JHSPH) developed by the Johns Hopkins Center for a Livable Future [42]. The food stores data included attributes for grocery stores, supermarkets, gas stations, and pharmacies. The GIS data of hospitals (acute, general, and special) licensed by the Maryland Department of Health and Mental Hygiene Office of Health Care Quality were obtained from Maryland's Mapping & GIS Portal [43]. Using these datasets, we calculated the sum of the number of food stores (grocery stores, supermarket, gas station, pharmacy), restaurants, pharmacies, and hospitals for each study MCD in Maryland. For California, the data on hospitals (2020) were obtained from Esri Data & Maps [38] and the data for pharmacies (2019) were obtained from OpenStreetMap [39]. The data for food retail establishments were not available for California in this study due to the lack of data for many CCDs across California.

We calculated statistics such as the first quartile (Q1), third quartile (Q3), mean, and minimum values of the daily percent changes in number of people moving between pairs of study MCDs (or CCDs) between 31 March and 24 April 2020 (i.e., after stay-at-home order) for Maryland and between 31 March and 19 April for California to characterize the mobility trends. Using linear regression analysis, we analyzed the relationships between the local vegetation level (i.e., EVI), presence of parks, and percent changes in mobility. A linear regression analysis was used for these statistics and parks, EVI, and urbanicity to examine if mobility patterns during the early stage of COVID-19 pandemic differed by green space (i.e., incorporating state parks or EVI level). We applied several different statistical models with different sets of confounders. The Q3 of mobility changes was used as a dependent variable as it represented the best normal distribution in Maryland (Supplementary Figure S1), while it was slightly skewed in California. We examined if results for green space and mobility trends were confounded by urbanicity (e.g., population, percent impervious area). We conducted regression analyses separately for each state.

## 3. Results

Descriptive statistics of the study regions are presented in Table 1. The higher average percent of impervious area in the MCDs in Maryland where the Facebook users' movement was observed

(32.7, SD = 29.3) compared to the MCDs for which the user's movement was not found indicate higher urbanicity level (Table 1). Average population density was higher in the MCDs where the mobility data were available. The range of EVI in the study MCDs in Maryland was narrower (0.15 to 0.29) compared to the range of EVI in the study CCDs in California (0.07 to 0.42). On the contrary, population density and percent impervious area were slightly higher in CCDs where the Facebook users' movement was not observed in California. The average of percent change in the number of users moving between pairs of the MCDs in a day and across the study MCDs in Maryland was −23.7 (SD = 21.5). The maximum reduction in number of users moved between MCDs in a day was −78.4% across all MCDs. In California, the average percent change in mobility between pairs of the CCDs in a day was −29.8 (SD = 25.7) along with the maximum percent change in mobility of −94.6%. The Q1, Q3, and median of percent changes in the number of travelling users indicate that mobility declined during the study period in most regions in both Maryland and California, although mobility did increase for some regions. The distance travelled between the study regions gradually decreased during the study period (Supplementary Figure S3).

Figure 1 represents the average daily percent changes in the number of users traveling into the study areas during the COVID-19 pandemic in Maryland and California. The trend showed a decreasing pattern from the beginning of the study period until the end of March and remained constantly at a low level until the end of the study period and the decrease in mobility was the highest in MCDs with the lowest level of EVI (i.e., <0.21) in Maryland. The decrease in mobility was the lowest in CCDs with the medium level of EVI (i.e., 0.24–0.29) in California.

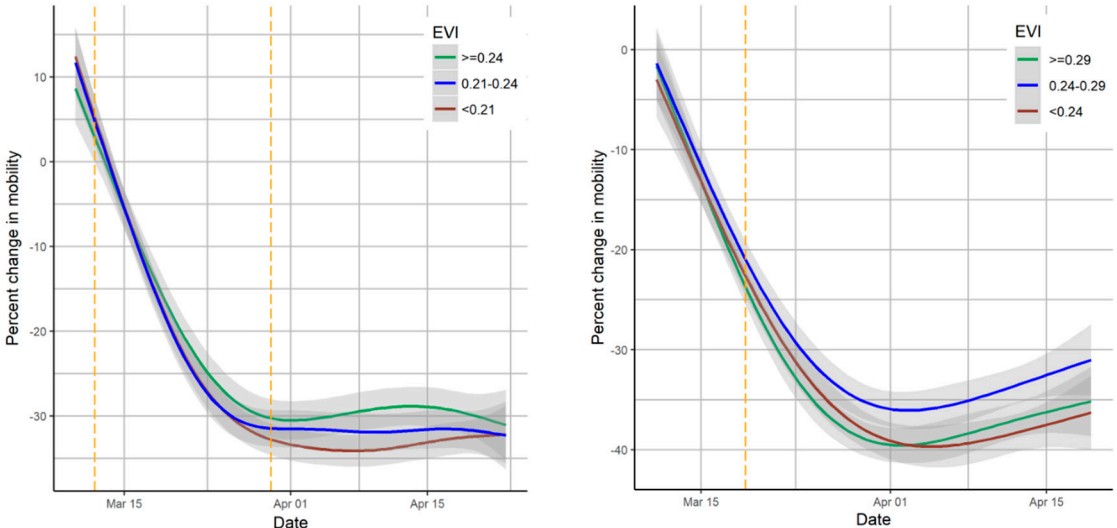

**Figure 1.** Average daily percent changes in mobility among study areas by EVI level in Maryland (**left**) and California (**right**) during the COVID-19 pandemic. Percent change in mobility is the percent change in the number of users traveling in the daytime (8 a.m.–4 p.m.) in a given day compared to the same time and the same day of the week in the reference period (26 February–10 March). Solid line: LOESS smoothing line; grey area: 95% confidence interval of LOESS line; yellow dotted lines for Maryland: declaration of state of emergency and the following stay-at-home order; yellow dotted line for California: stay-at-home order.

**Table 1.** Descriptive statistics of vegetation level and mobility trends during the COVID-19 pandemic by minor civil division (MCD) in Maryland (11 March–24 April 2020) and census country division (CCD) in California (11 March–19 April 2020).

| Variable | MCDs Where Mobility Was Observed | | | | | MCDs Where Mobility Was Not Observed | | | | |
|---|---|---|---|---|---|---|---|---|---|---|
| | Mean (SD) | Min-Max | Q1 | Q3 | Median | Mean (SD) | Min-Max | Q1 | Q3 | Median |
| **Maryland** | | | *n* = 76 | | | | | *n* = 154 | | |
| EVI (range −1 to 1) | 0.22 (0.03) | 0.15–0.29 | 0.19 | 0.23 | 0.22 | 0.25 (0.04) | 0.05–0.31 | 0.23 | 0.27 | 0.25 |
| Percent of impervious area (%) | 32.7 (29.3) | 0.0–92.3 | 7.2 | 61.8 | 19.0 | 1.3 (3.5) | 0.0–33.3 | 0.0 | 1.1 | 0.1 |
| Population (persons) | 61,080 (74,559) | 161–566,200 | 25,810 | 84,830 | 44,920 | 8892 (8860) | 436–39,470 | 2224 | 13,030 | 6319 |
| Population density (persons/km$^2$) | 805.3 (901.9) | 0.4–6062.0 | 218.4 | 991.4 | 563 | 90.5 (100.6) | 3.1–600.8 | 21.3 | 132.0 | 58.2 |
| Number of food retail and hospitals * | 187 (321) | 2–2712 | 58 | 206 | 126 | | | | | |
| Percent change in mobility (%) | −23.7 (21.5) | −78.4–120.7 | −38.9 | −10.8 | −26.3 | | | | | |
| Travel distance (km) † | 7.3 (3.4) | 3.8–23.2 | 5.3 | 8.4 | 6.8 | | | | | |
| Presence of parks * | Number | % | | | | Number | % | | | |
| Yes | 65 | 85.6 | | | | 123 | 79.9 | | | |
| No | 11 | 14.4 | | | | 31 | 20.1 | | | |
| **California** | | | *n* = 241 | | | | | *n* = 156 | | |
| EVI (range −1 to 1) | 0.26 (0.07) | 0.07–0.42 | 0.22 | 0.31 | 0.27 | 0.27 (0.07) | 0.07–0.31 | 0.22 | 0.32 | 0.27 |
| Percent of impervious area (%) | 10.4 (19.7) | 0.0–100.0 | 0.2 | 8.3 | 1.3 | 12.7 (23.7) | 0.0–100.0 | 0.4 | 10.7 | 2.2 |
| Population (persons) | 110,587 (294,124) | 741–2,457,972 | 7026 | 74,510 | 20,785 | 67,965 (174,581) | 262–1,664,311 | 5531 | 59,500 | 12,750 |
| Population density (persons/km$^2$) | 247.5 (584.5) | 0.1–5136.7 | 5.6 | 166.0 | 34.1 | 303.7 (680.3) | 0.1–4951.0 | 13.9 | 221.7 | 49.3 |
| Number of hospitals and pharmacies | 6.1 (19.0) | 0.0–190.0 | 0.0 | 5.0 | 1.0 | 3.4 (9.7) | 0.0–98.0 | 0.0 | 3.0 | 1.0 |
| Percent change in mobility (%) | −29.8 (25.7) | −94.6–377.9 | −47.2 | −16.0 | 34.3 | | | | | |
| Travel distance (km) † | | | | | | | | | | |
| Presence of parks * | Number | % | | | | Number | % | | | |
| Yes | 216 | 89.6 | | | | 118 | 75.6 | | | |
| No | 25 | 10.4 | | | | 38 | 24.4 | | | |

* Food retail establishments and hospitals refer to grocery stores, supermarkets, gas stations, pharmacies, restaurants, and hospitals. The 'presence of parks' variable refers to national parks and forest, state parks and forests, hiking trails, and local-scale parks. † The average distance between the centroids of spatial grid cells (Bing tile) the users traveled between for each day during the study period. EVI = Enhanced Vegetation Index.

Figure 2 shows the descriptive statistics (Q1, Q3, median, and average) of daily percent changes in mobility and EVI values at the MCD level (or CCD level) of the study regions. The scatter plots showed that the regions with high EVI values may have lower reduction in their mobility trend during the study period in Maryland and California. However, the correlation coefficients for the EVI in Maryland were 0.11, −0.01, 0.08, and 0.05 for the Q3, Q1, mean, and median of mobility changes, respectively, indicating no significant correlations. Similarly, for California, no significant correlations were observed for EVI and statistics of mobility changes (0.07, 0.12, 0.08, and 0.08 for the Q3, Q1, mean, and median of mobility changes, respectively).

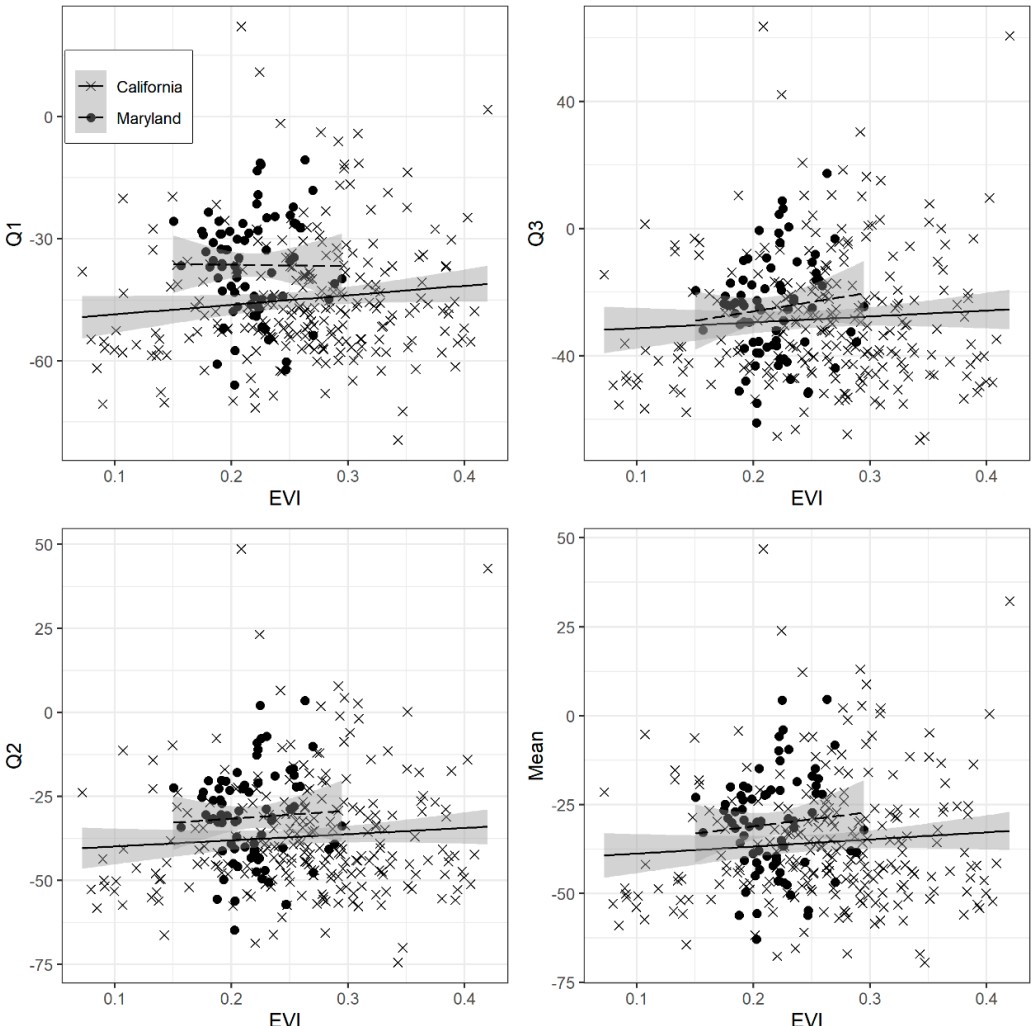

**Figure 2.** Scatter plots of statistics of percent changes in mobility and EVI in the study areas in Maryland (*n* = 76) and California (*n* = 241). The grey area is 95% confidence intervals for linear regression lines.

Figure 3 presents the locations of parks and forest and the geographical patterns of mobility changes (Q3) and EVI. On average, the sizes of parks and forests as well as the size of county subdivisions were larger for California than Maryland. The movement of users was mostly observed in the central areas of Maryland including MCDs adjacent to Baltimore, Maryland. The geographical pattern of EVI showed a relatively particular pattern with higher EVI values for central western parts in Maryland and western parks of California, while the geographical patterns of mobility changes appeared to be random in study regions in both states.

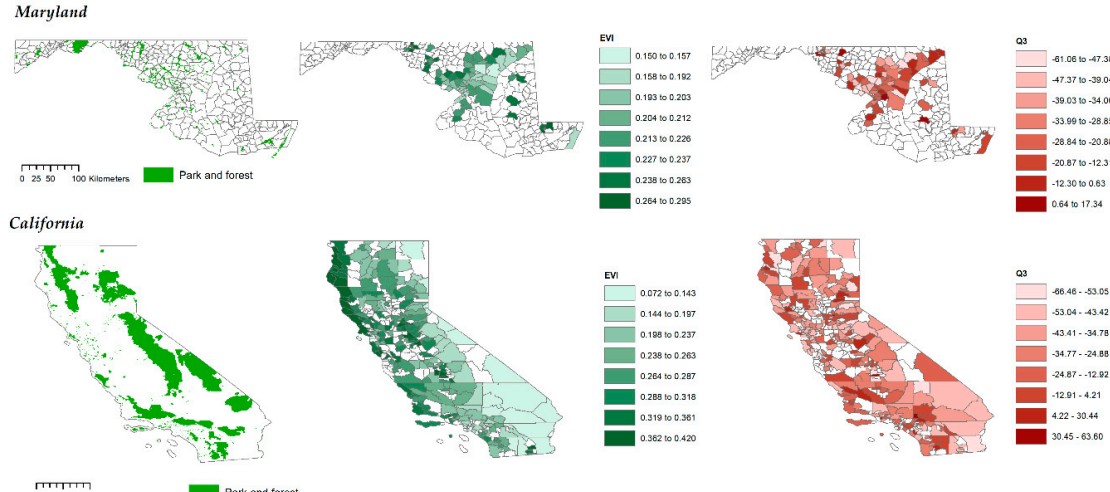

**Figure 3.** Location of parks and forest and the spatial patterns of EVI and the statistics (Q1, mean, Q3) of percent changes in the number of users moving into each subdivision in Maryland (top) and California (bottom) during the study period (11 March–26 April 2020). Blank area: County subdivisions (MCDs, CCDs) where users' movement data were unavailable.

Figure 4 presents the mobility change and EVI for MCD (CCD) groups with and without parks. MCDs with parks in Maryland showed slightly lower reduction in mobility compared to MCDs without parks. In California, reduction in mobility was relatively lower in CCDs with parks compared to CCDs without parks. EVI was lower in MCDs and CCDs with parks in Maryland and California.

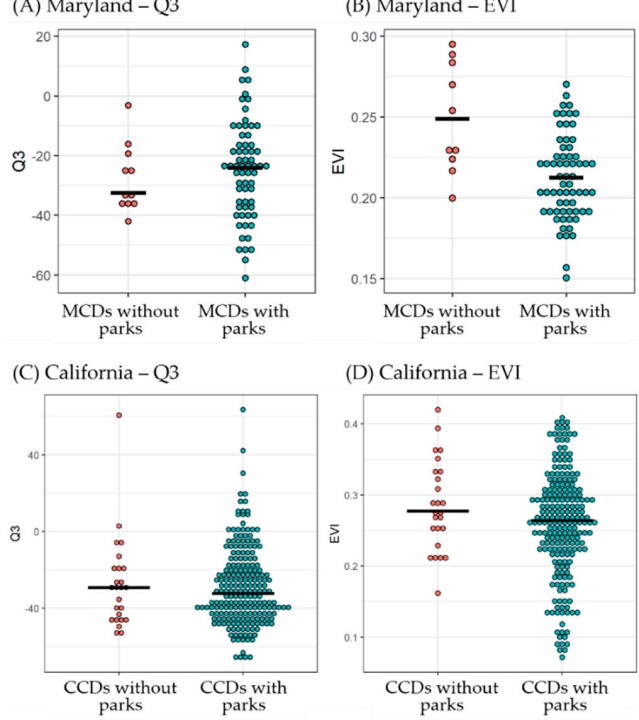

**Figure 4.** Statistics of percent changes in mobility (Q3) and EVI for MCD (CCD) groups with and without parks in Maryland and California. Solid lines are median.

Regression coefficients of EVI from the linear regression analysis are presented in Table 2. Although MCDs (CCDs) with high EVI values tended to have lower reduction in their mobility (Figure 2), EVI at the MCD (CCD) level in Maryland and California were not significantly associated with mobility changes (Q3) during the study period in any model.

**Table 2.** Regression coefficients of EVI for the relationship with mobility changes (Q3) in the study county subdivisions (MCD, CCD) in Maryland and California.

| Region | Model 1 | | | Model 2 | | |
|---|---|---|---|---|---|---|
| | Beta | (95% CI) | *p*-Value | Beta | (95% CI) | *p*-Value |
| Maryland (*n* = 75) | −1.06 | (−2.49, 0.38) | 0.154 | −1.31 | (−2.63, 0.02) | 0.067 |
| California (*n* = 241) | 0.02 | (−9.02, 9.74) | 0.933 | 0.13 | (−0.23 0.50) | 0.473 |

Note: EVI = Enhanced Vegetation Index. Model 1 was adjusted for presence of parks (all types), log population density, and number of food retail establishments and hospitals; Model 2 was adjusted for presence of parks (all types), percent impervious area, and number of food retail establishments and hospitals.

Results of the linear regression analysis for presence of parks are shown in Table 3. Presence of state parks in MCD boundary in Maryland was significantly associated with lower reduction in mobility (7.62, 95% CI: 0.28, 14.97) in Model 1, whereas Model 2 and Model 3 did not show significantly lower reduction in mobility. When percent impervious area was included as adjustment instead of population density, having state parks or EVI did not show significant effects on mobility changes. In the third model incorporating the variable of presence of state parks, number of food retail establishments and hospitals, and population density, presence of state parks showed significant effects on mobility changes at a 0.10 significance level. All types of parks and local-scale parks showed no significant relationship with mobility changes in Maryland during the study period. On the other hand, presence of local-scale parks in CCDs in California showed significantly lower reduction in mobility in Model 1 (7.03, 95% CI: 0.12, 13.94) and Model 3 (7.02, 95% CI: 0.13, 13.92). Mobility reduction tended to be lower in CCDs with any type of parks or state or national parks but the results were not significant in California.

**Table 3.** Results of linear regression analysis on the mobility changes (Q3 and presence of parks in the study county subdivisions (MCD, CCD) in Maryland and California.

| Variable | Model 1 | | | Model 2 | | | Model 3 | | |
|---|---|---|---|---|---|---|---|---|---|
| | Beta | (95% CI) | *p*-Value | Beta | (95% CI) | *p*-Value | Beta | (95% CI) | *p*-Value |
| Maryland (*n* = 75) Presence of parks by type (ref = having no park) | | | | | | | | | |
| Total | 7.31 | (−4.97, 1.96) | 0.248 | 1.57 | (−9.96, 13.10) | 0.790 | 7.00 | (−4.61, 18.60) | 0.242 |
| State and National | 7.62 | (0.27, 14.97) | 0.045 | 5.56 | (−1.87, 13.00) | 0.145 | 6.04 | (−1.05, 13.13) | 0.100 |
| Local | 5.79 | (−5.21, 16.80) | 0.306 | 0.057 | (−9.52, 10.65) | 0.913 | 5.71 | (−4.87, 16.29) | 0.294 |
| California (*n* = 241) Presence of parks by type (ref = having no park) | | | | | | | | | |
| Total | 0.36 | (−9.02, 9.75) | 0.939 | −0.04 | (−9.71, 8.87) | 0.930 | 0.33 | (−9.00, 9.66) | 0.945 |
| State and National | 0.22 | (−6.20, 6.65) | 0.946 | 0.38 | (−5.91, 6.68) | 0.906 | 0.20 | (−6.19, 6.58) | 0.952 |
| Local | 7.03 | (0.12, 13.94) | 0.048 | 5.17 | (−1.01, 11.36) | 0.102 | 7.02 | (0.13, 13.92) | 0.047 |

Note: Model 1 was adjusted for number of food retail establishments and hospitals, population density, and EVI; Model 2 was adjusted for number of food retail establishments and hospitals, percent impervious area, and EVI; Model 3 was adjusted for number of food retail establishments and hospitals and log population density.

## 4. Discussion

We examined if the amount of green space such as parks and vegetation level are associated with human mobility under pandemic mitigation policies due to COVID-19 in Maryland and California. A novel result of this study is that the decline of mobility during the COVID-19 pandemic appeared to be lower in regions with green space such as state parks in Maryland and local parks in California. This may imply that people sought green spaces or beach areas to cope with the stress of the pandemic and to perform outdoor activities with social distancing. Further, our results demonstrated that types and scales of parks (e.g., state, local) may impact the effects of parks on mobility changes in different US states potentially due to differences in size of administrative regions (e.g., county subdivision), population density, land cover, or scales of green space (e.g., parks, forest, beach). Our study does

not provide data for an association between mobility and COVID-19 transmission. Nonetheless, the potential impact of green space on mobility during a pandemic shown in our study and other recent literature [7,31,32] implies the need for future design and planning for green space and open spaces where disease control measures such as physical distancing can be performed during an outbreak [31,44,45]. Our results suggest that different plans for sheltering in place in relation to local built environment at regional levels are required. Furthermore, this study is in the scope of sustainable development for resilience and preparedness with nature against future pandemics [46]. Currently, there is an urgent need for studying how mental health consequences can be mitigated in a pandemic [24]. Previous work suggested that a long quarantine duration can be a major stressor causing emotional distress and increased risk of psychiatric illness, unhealthy behaviors, and noncompliance with mandate public health guidelines [47]. A study conducted in Italy provided evidence that the COVID-19 pandemic was significantly associated with increased risks of developing depression, anxiety, and sleeping disorders during lockdown [48]. Given the impact of the pandemic on mental health, and the benefits of green space to mental health, understanding the relationship between the pandemic and green space is paramount.

Many studies have utilized the vegetation index to assess amount of local green space in relation to provision of green space [49,50] and health effects of green space [51]. Given the negative correlation between EVI and urbanization (e.g., population density, impervious areas) in our study, the differences in mobility reduction by EVI levels reflect that people traveled less to more populated areas for safety. This was seen for the graph of daily mobility changes by EVI levels in California. Our data also showed that the presence of parks did not corresponded with higher vegetation level at the MCD (CCD) level in Maryland and California, which indicates that parks were more located in urbanized areas. To control for the effect of urbanization, we examined two types of urbanization indicators (i.e., population density, percent impervious area) in our statistical models. Our results found that the presence of parks and other types of natural areas for recreational activities (e.g., fishing, hunting) was more significantly associated with mobility patterns than was vegetation level. This was particularly due to the characteristics of land cover in CCDs in California where state and national parks were located. These large-scale parks were located in CCDs in central eastern parts of California, where there are also deserts such as the Mojave Desert bounded by the Tehachapi Mountains, San Gabriel and San Bernardino Mountains. These areas are also bounded by Arizona and Nevada where the land cover is largely desert at the borders with California. Due to these characteristics, EVI tended to be lower in these areas despite the presence of state and national parks. On the other hand, lower effects of EVI could be related with the nature of the vegetation index as it does not fully represent the quality of greenness or the actual accessibility (e.g., presence of entry points for people) to green space, although the vegetation index has been used extensively to measure greenness [52]. Study results for the best greenness metrics for assessing health effects are inconsistent. Thus, we suggest that future studies on green space and health consider types and volume of green space in addition to the vegetation index to better understand the functions of green space for outdoor activities and in relation to health, and that research on the characteristics of green space that are most relevant for health be conducted.

Studies suggested that transmission of COVID-19 can be mitigated with further local control measures including suspending public transport, closing entertainment venues, and banning public gatherings in addition to travel bans [1,53]. In response to the pandemic, the Maryland Department of Natural Resources postponed and/or canceled programs and events and closed areas in parks where the public may congregate, such as visitor centers, administrative buildings, and shelters [54]. Several state parks and golf courses were closed to the public during our study period. Gatherings of more than 10 persons were prohibited, and state residents were encouraged to keep distance when outside, not participate in team sports, and avoid touching surfaces that may be handled by others (e.g., playground equipment and benches). Access to most parks in Maryland as of April 2020 was not restricted due to COVID-19. Parks and trails remained open for activities such as hiking, biking, or walking, although state park beach areas were closed on 30 March [55]. Thus, we assumed that

the closure of some parks due to COVID-19 did not affect our analysis on mobility and green space, although uncertainty remains. Similarly, California Department of Parks and Recreation announced its first order for suspension of tours to some California State Parks in 14 March in an effort to protect public health from COVID-19 and announced additional temporary full closure of state parks in 3 April [56]. Since then, access to California State Parks has continuously changed according to the severity of disease spread and compliance with state and local public health ordinances of communities. It is important to note that having different regulations for access to green space hinders direct comparisons of the relationships between green space and human mobility among different US states. This may justify analysis for individual states or administrative units with similar scales.

While our study focused on the increases in visits to green space at an early stage of the pandemic, which was also identified in other literature [7,31,32], COVID-19 has deteriorated the ability of people to utilize urban green space resources (e.g., parks, playgrounds) in many communities [57]. Hall et al. suggested that efforts should be made to assess the lasting effects of pandemics on physical activities along with closures of open green spaces [57]. In addition to such call for research, our empirical results on the relationships between green space and human mobility suggest the need for further understanding and study directions about the way people use green space. Although crowding in urban parks is likely to contribute to transmission of COVID-19, timely data for the number of visitors in green space are not readily available. Further, it is unknown if and how people perform safety measures such as wearing masks and practicing physical distancing in public green spaces. While numerous studies have provided evidence of health benefits of green space on psychological and physical health through stress relief, enhanced physical activities, and social cohesion [25,27], there is a lack of information on which benefits occur when people interact with green space during the pandemic with social distancing policies, and how these relate to the health detriments of increased disease transmission. It is unclear if the frequency or purposes of visiting green space have changed due to the COVID-19 pandemic and, if so, how those changes and resultant health effects differ by subpopulation (e.g., race/ethnicity, socioeconomic status). Future studies will need to address the complex relationships among COVID-19, greens space, and health effects in order to better understand how the pandemic affects our interaction with green space and its health impacts.

Our study has several strengths. To our best of our knowledge, this study is the first to examine the effect of amount of green space on human mobility during the COVID-19 pandemic in Maryland and California. We used novel social media app-based mobility data and considered the density of basic social assets (e.g., number of food retail establishments and hospitals) within county subdivisions (MCDs, CCDs) as an indirect indicator of people's travel outside county subdivisions for basic services (e.g., grocery shopping, health care services). Our results provide timely information relevant for the policies of control measures of a pandemic in relation to green spaces.

This study has some limitations. The mobility identified based on app-based location services can only be observed when the app is active and is thereby may be affected by situations for which people are likely to use a phone (e.g., searching directions, connecting with friends, posting photos). We were not able to consider user's movement within a Bing tile. We could not consider all county subdivisions in Maryland and California due to the lack of mobility data in some subregions based on the privacy protection and lack of Facebook app users. Facebook Data for Good provides daily mobility changes, only since March 2020 with February as the reference period, which prevents comparisons of mobility patterns across years (e.g., the same day one year before) and prevents analysis of mobility patterns during the pre-pandemic period. As a result, we could not disentangle mobility changes due to the state-of-emergency declaration from a potential seasonal pattern of mobility though February–April. A recent study argued that seasonality rather than the COVID-19 pandemic caused the reported increases in park visitations in Google Mobility Reports for some Western US counties [58]. We note that our study area likely shares the same seasonality of mobility across county subdivisions, therefore, the potential impact of seasonality does not likely explain differences in mobility across study subregions. We also note that our main question was the impact of green space on mobility

patterns during the COVID-19 pandemic rather than the impact of the state-of-emergency declaration or specific policies or guidance, which may vary locally, on mobility patterns during the lockdown period. Another limitation is that we could not consider Baltimore city in our regression analysis for Maryland as it was an extreme outlier in terms of population size and number of food retail establishment and hospitals compared to other MCDs, which affected our assumption for the linear relationships for mobility changes and the covariates in the model. Although we used commonly applied datasets for food retail establishments in Maryland, these measures do not include every type of food supply store (e.g., farmer's markets, food trucks). Therefore, there is an uncertainty regarding weather such results would be robust when the choices of types of food retail or health care facilities are different. In addition, food retail data were not available for California in this study. Thus, a cautious interpretation is required of the study results and their comparisons between the study states. We note that our findings of the relationship between presence of parks and mobility changes during the study period may have limited generalizability beyond the sub-urban or peripheral urban areas. This leaves a question for the impact of urbanicity on the use of green space and requires future studies comparing highly urbanized cities in the US. Only Maryland and California were examined in our analysis. The Facebook Data for Good has gradually published open data for mobility for more expended regions since February 2020. However, those updated mobility data have a discrepancy for observation period with previous datasets. We considered Maryland and California, which had the same reference period for daily estimating mobility changes across the US.

Due to the unprecedented nature of the pandemic and the critical need for scientific evidence, many research studies related to COVID-19 have been conducted in an urgent manner in spite of methodological challenges, which leads to some degrees of uncertainty. For the sake of public health against the novel impact of COVID-19, our study examined a unique hypothesis for the relationship between green space, COVID-19, and mobility by combining social media data and satellite remote sensing technology. However, future work is needed to confirm the findings presented here and to investigate relevant questions. Here we discuss suggestions for future studies. First, higher resolution of mobility data would aid understanding of neighborhood-level mobility patterns and their relationships with green space. Individual-level data could illuminate differences in patterns by population characteristics, such as by socioeconomic status and race/ethnicity, which is important given the higher health burden faced by these groups. However, ethical issues of privacy should be considered and continuous efforts to find the best methodological approach for mobility data with lower spatial resolution and with individual-level data. Discussions on finding a scientifically and socially acceptable trade-off between privacy and scientific data needs might support future research. Second, there are various types of mobility data available. While Facebook Data for Good only provides mobility data of Facebook users, mobility data are obtainable through mobile phone networks (i.e., cellular networks). Mobile phone networks are composed of geographic zones (called 'cell') around a phone tower and each mobile phone can be located by identifying the geographical location and the associated cell of its transmitting phone tower [59]. This type of mobility data will incorporate any people who use mobile phone services for a given geographical zone. Facebook's mobility data still have strengths of researcher-friendly pre-generated outcomes by the Facebook Data for Good team, whereas the data based on mobile phone networks from phone towers may require the researcher's own computation logics and detailed understanding of such data. Another type of mobility data is specifically available for "point-of-interest (POI)." Mobility to POI data can be considered in assessing the mobility patterns to each destination of green space including parks and they may be less sensitive to challenges from low spatial resolutions of mobility estimation. Some datasets are freely shared for research through an existing collaborative consortium between the data company and research organizations. Considering this type of datasets based on POI would be helpful to understand the health effects of mobility to green space. Third, there are inconsistencies in the definition of green space among different countries and disciplines, and while various datasets on green space provide information, none portray the rich characteristics of heterogeneity in green space such as different

types of vegetation, park access, park features, etc. Differences in scales, features, and quality in green and open space may result in difficulties in comparing study results among different regions and could obscure important relationships between mobility and green space. It is imperative to understand sizes of geographic divisions, environmental characteristics (e.g., land cover), green space features (e.g., parking availability), and urbanization in study regions to investigate the effects of green space on human mobility. Fourth, more information should be produced for human behavior patterns for using green space. The pattern of using green space (e.g., purposes of visiting) would vary among communities by urbanicity, culture, or safety level, but less is known for these patterns. Studies, possibly with survey of local residents, would be helpful for future research. In spite of these limitations, our work aims to inform future studies of urban sustainability and public health in relation to green space under the threats of global pandemic.

## 5. Conclusions

The pandemic has likely changed our relationship with green space, which has numerous established public health and societal benefits, and the nature of this change is not fully understood. Scientific evidence is needed regarding travel to green space, an important environmental determinant of human health, under normal conditions and pandemics. We investigated the effect of green space on human mobility patterns during the COVID-19 pandemic using large-scale mobility data from social media; our findings imply potential increases in usage of green space in Maryland and California, USA, particularly parks and environment for recreational activities when other essential social activities were prohibited or discouraged due to the control measures of COVID-19. Results suggest that understanding environmental factors associated with mobility changes during a pandemic can aid decision-makers with preparing preventive measures against public health burdens caused by pandemics. We urge future studies to explore these findings, expand relevant data sources, investigate methodologies to estimate the health benefits of green space impeded by a pandemic and the most vulnerable persons to such damages.

**Supplementary Materials:** The following are available online at http://www.mdpi.com/2071-1050/12/22/9401/s1, Figure S1: Histograms of the statistics of the daily percent changes in mobility in Maryland, Figure S2: Histograms of the statistics of the daily percent changes in mobility in California, Figure S3: Trend in the distance traveled by users during the study period, Figure S4: Pair-wise scatter plots for EVI and covariates in Maryland, Figure S5: Pair-wise scatter plots for EVI and covariates in California, Table S1: Regression coefficients of covariates in the statistical models including parks (all types), EVI, retail and hospitals, population density, and percent impervious area.

**Author Contributions:** Conceptualization, S.H.; methodology, S.H.; software, S.H.; validation, S.H. and M.L.B.; formal analysis, S.H.; investigation, S.H.; resources, S.H.; data curation, S.H.; writing—original draft preparation, S.H. and C.C.L.; writing—review and editing, S.H. and M.L.B.; visualization, S.H. and M.L.B.; supervision, M.L.B.; project administration, M.L.B.; funding acquisition, M.L.B. All authors have read and agreed to the published version of the manuscript.

**Funding:** This research was funded by the U.S. Environmental Protection Agency, grant number RD835871.

**Acknowledgments:** This publication was developed under Assistance Agreement No. RD835871 awarded by the U.S. Environmental Protection Agency to Yale University. It has not been formally reviewed by EPA. EPA does not endorse any products or commercial services mentioned in this publication. The content is solely the responsibility of the authors and does not necessarily represent the official views of the EPA. Authors appreciate Facebook for providing Facebook Disease Prevention Maps (https://dataforgood.fb.com/tools/disease-prevention-maps/).

**Conflicts of Interest:** The authors declare no conflict of interest.

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
