# Peer review of "Relationships between Local Green Space and Human Mobility Patterns during COVID-19 for Maryland and California, USA"

_sustainability, doi:10.3390/su12229401_

Round 1

Reviewer 1 Report

The topic and the objectives are relevant to the research agenda on the contribution of nature to human health and wellbeing. I think that a more focused and exhaustive introduction would help to correctly identify the relevance and potential uses of this study. The availability of data and soundness of the relationshp between lockdown mesures, movements, and greenspace is still scarce and at its beginning. In my opinion this is still a slippery territory for research. Your study contributes in finding methodologyes to invstigate the topic, but more extensive studies are needed so to have impactfull results and conclusions.

Few comments for improving the cohesion of the presentation:

Line 50 – few other studies… these studies are supporting the same results as the nes in the line before, right? Make it more clear

Line 59 – sustainable capacity? Not clear the meaning

Line 62 – After this sentence the reader feels the need to understand “and what about green space influence on mobility on normal conditions?”. Assessing the influence of a factor under extreme condition without considering the business-as-usual could be a bias for the interpretation of results. The relationship between mobility and COVID is well discussed, as for the one between COVID and mental health. Weak discussion for mobility and green links, and how to connect in practice your results with improvement of mental health.

Lines 68-82 you give a lot of space to another piece of this complex dynamics, namely the potential of nature space of mitigating stress and anxiety. Since you cannot prove this link with your research, I would suggest reducing this part in the introduction and use it in the discussion of the results instead. I would simplify as much as possible and focus on solid research and variables of the relationship you want to investigate.

Line 149 Please make clear, a brief sentence is enough, the choice of selecting food retails and hospitals. And discuss more the consequence of this choice

Line 236 - Sorry if I missed this point, but could be that where the EVI is higher the population density is lower and areas are less urbanized? This may imply that people are let affected by the social distancing measures and more keen to move safely; more used to exercise in the green?

The discussion is well articulated, highlighting all the critical aspects and future researches. We completely understand the effort and the barriers in doing at this stage a research on mobility. Maybe try to stress out better the practical implication of your research and the possibility for scaling-up your methodology.

Thank you very much

Author Response

The topic and the objectives are relevant to the research agenda on the contribution of nature to human health and wellbeing. I think that a more focused and exhaustive introduction would help to correctly identify the relevance and potential uses of this study. The availability of data and soundness of the relationship between lockdown measures, movements, and greenspace is still scarce and at its beginning. In my opinion this is still a slippery territory for research. Your study contributes in finding methodologies to investigate the topic, but more extensive studies are needed so to have impactful results and conclusions.

A: We modified the introduction as recommended and based on comments from reviewers. We have added text to address uncertainties regarding this work and the need for future studies. Please see our specific responses to comments below.

Few comments for improving the cohesion of the presentation:

Line 50 – few other studies… these studies are supporting the same results as the nes in the line before, right? Make it more clear

A: The studies in this line has the same context with those in the previous sentence. We revised line 55 from “A few studies…” to “A few other studies….”.

Line 59 – sustainable capacity? Not clear the meaning

A: Thank you for this comment. We agree that the text was unclear. We revised the sentence from “The COVID-19 pandemic along with other major environmental crisis such as climate change shed a light on a need for better understanding about the nature as a sustainable capacity and resilience to deal with complex health crisis” to “The COVID-19 pandemic along with other major environmental crisis such as climate change shed a light on the need for better understanding about how to promote resilience and capacity of societies to deal with complex  health crises” (lines 77-79).

Line 62 – After this sentence the reader feels the need to understand “and what about green space influence on mobility on normal conditions?”. Assessing the influence of a factor under extreme condition without considering the business-as-usual could be a bias for the interpretation of results. The relationship between mobility and COVID is well discussed, as for the one between COVID and mental health. Weak discussion for mobility and green links, and how to connect in practice your results with improvement of mental health.

A: Thank you for this comment. We note that the analysis only for the period under pandemic is a limitation of this work. However, as discussed in the limitation of our study, the mobility data used in this work does not provide mobility information for the time period before the COVID-19 pandemic. Thus, we could not provide the analysis for the influence of green space on mobility under normal (non-pandemic) conditions. Our analysis does look at time periods early in the pandemic (March on) and reflects changes in travel to green space. We added text to note the limitation due to data availability and to highlight the need for future research to confirm and further explore these findings (lines 419-420). We also expanded our study area from the original study of Maryland to include California as well, based on data that is newly available since our original submission.

We  added a new sentence of “Given the restrictions on the gathering of people, particularly in indoor settings, during COVID-19, understanding changes in the use of green space contributes to our understanding of how green space relates to the ability of communities to cope with the stress from quarantine and pandemic, such as by playing a role as an alternative place for physical activity.” (line 84-87).  

Lines 68-82 you give a lot of space to another piece of this complex dynamics, namely the potential of nature space of mitigating stress and anxiety. Since you cannot prove this link with your research, I would suggest reducing this part in the introduction and use it in the discussion of the results instead. I would simplify as much as possible and focus on solid research and variables of the relationship you want to investigate.

A: We have revised the third paragraph of Introduction. We removed some sentences explaining previous findings retarding mental health and green space and combined the references for a single sentence (i.e., "Several studies argued that physical inability as ....")(Lines 70-75). As suggested, and in line with other reviewer comments, we moved some of this discussion and references to the Discussion section instead (lines 328-336).

Line 149 Please make clear, a brief sentence is enough, the choice of selecting food retails and hospitals. And discuss more the consequence of this choice

A: Thank you for this suggestion. We added a new sentence to the Discussion to note that the potential uncertainty of the results based on the choices of food retail establishments and hospitals. Please see lines 434-436.

Line 236 - Sorry if I missed this point, but could be that where the EVI is higher the population density is lower and areas are less urbanized? This may imply that people are let affected by the social distancing measures and more keen to move safely; more used to exercise in the green?

A: We agree with this comment. Our analysis showed negative correlations between EVI and urbanization variables. To better examine the relationship between green space and mobility patterns, we further considered the presence of state parks and nature areas in addition to EVI, to better characterize the green space beyond the measure of EVI. For this revised paper, we added data on local parks as well. To note this point more clearly, we added a new sentence of "Given the negative correlation between EVI and urbanization (e.g., population density, impervious areas), the differences in mobility reduction by EVI levels shown in our results might reflect that people traveled less to more populated areas for the sake of safety." in Discussion (line 338-341).

The discussion is well articulated, highlighting all the critical aspects and future researches. We completely understand the effort and the barriers in doing at this stage a research on mobility. Maybe try to stress out better the practical implication of your research and the possibility for scaling-up your methodology.

A: We appreciate this suggestion. By addressing approaches to scale up this type of study, we think the manuscript has been significantly improved. Please see the last paragraph of Discussion where we discuss methodological challenges and specific suggestions for future studies.

Thank you very much

Reviewer 2 Report

The paper Relationships between Local Green Space and Human Mobility Patterns during COVID-19 for Maryland, USA raises very important issue related to human mobility during recent pandemia. I suggest some improvements to raise paper quality.

  • The aim of the paper has to be added in the abstract
  • There are hypothesis in the introduction but the aim of the paper must be described
  • mathematic formulas should be added in chapter 2
  • The paper is rather short. The paper will be more readable if suplementary graphs will be added into main text and analyzed there.
  • Figures in Results chapter should be wider analyzed

good luck!

Author Response

The paper Relationships between Local Green Space and Human Mobility Patterns during COVID-19 for Maryland, USA raises very important issue related to human mobility during recent pandemic. I suggest some improvements to raise paper quality.

The aim of the paper has to be added in the abstract

A: We revised the abstract to add a new sentence specifying our aim. Please see “The aim of this study was to investigate the effect of green space on mobility reductions during the early stage of COVID-19 pandemic in Maryland and California, USA.” in lines 11-12.

There are hypothesis in the introduction but the aim of the paper must be described

A: Thank you for this comment. We revised the last 3 paragraphs of the Introduction to more clearly explain the study background and our study’s aim as well as the hypothesis. A new sentence “The aim was this study is to investigate green space as a potential factor influencing mobility patterns during a pandemic.” was added.

Mathematic formulas should be added in chapter 2

A: The equation of the percent change of movement was added to lines 124-126. The equation for NDVI was added to Line 148.  

The paper is rather short. The paper will be more readable if supplementary graphs will be added into main text and analyzed there.

A : To respond to this comment, we have moved some supplementary materials to the main results to expand the main paper. Further, we conducted additional analysis for extended study regions. Please note that California has been added to our study regions and the analyses was repeated for this state. The results were robust to these extended study regions (i.e., Maryland and California). With these new results, we increased the length and implications of our manuscript significantly. One additional table (Table 2) and a figure (Figure 3) were added to Results.

Figures in Results chapter should be wider analyzed

A: With the extended study regions, we produced new figures. Please see that California has been added to Figure 1, 2, and 3. Specifically, maps of parks in Maryland and California were added to the Results. Also, EVI and statistics of mobility changes in California were added.  Maps of locations of parks and EVI support the relationships between EVI and presence of parks shown in Figure 1 and Figure 4. We also expanded the text regarding the results.

Reviewer 3 Report

I found this paper quite well written. Some wording mistakes but nothing that a good check would miss.

My concern is with the low level of significance of the results presented. The topic is certainly of interest and timely (may be too much). 

The results are marginally significant and the indicators of Green Infrastructures quite crude. 

The use of Facebook data for Good could be an interesting source but the too coarse resolution makes the analysis pretty weak. 

Given this paper relies on free to access data and remote sensing, it would make sense to increase its robustness by enlarging considerably the study area. Why focusing only on a part of Marynland?

It could be interesting to compare areas with different regulations regarding access to Outdoor settings. 

The EVI is one possible index, having a better analysis regarding the amount of Green infrastructure would be required, not just pooling everything.

At this stage, I found this study too preliminary for publications. 

The conclusions are only weakly supported by observations. Combining FB data with people enquiries regarding their behaviour during the restrictions could also strengthen considerably the analysis.

Author Response

I found this paper quite well written. Some wording mistakes but nothing that a good check would miss.

A: We reviewed the paper for wording issues. Thank you for noting this issue.

My concern is with the low level of significance of the results presented. The topic is certainly of interest and timely (may be too much). 

A: Thank you for this comment. We agree that research related with COVID-19, including this work, have uncertainty. However, given the societal and health impact of the pandemic, such work needs to be conducted in an urgent and timely manner. We think that it is important to directly examine important research questions using available data for public health under this circumstance of an unprecedented disease outbreak rather than waiting for the relevant data to be complete, as decisions regarding response to the pandemic are taking place now. Still, we agree with the reviewer that limitations remain. Currently, the publicly available mobility data are still limited for the US. They have been randomly available in terms of locations and observation periods. Despite this challenge, for our revised study, we extended our analysis for California in addition to Maryland and our manuscript has been revised accordingly. The main findings for the effects of green space were robust. Please see the revisions marked throughout our manuscript. We also added text to highlight the reviewer’s point and note the various uncertainties and limitations of this work, the need for future work to confirm these findings, and specific topics for further study (lines 447-488).

The results are marginally significant and the indicators of Green Infrastructures quite crude. 

A: We agree that EVI does not fully capture the full characteristics of green infrastructure. We address this comment in two ways. First we considered additional data for different types of parks, which provides more context to the nature of green space. In the original version, we only had information for ‘state parks’, whereas for this version we obtained and incorporated information for local-scale parks as well.  The effects of parks were analyzed for our extended study regions (i.e., Maryland and California). Further, we added analysis to show how presence of green space spatially overlapped with land cover of our study regions, which provides information for the relationship between EVI and presence of green space. Please see Figure 1 and 3 for these new results.

The use of Facebook data for Good could be an interesting source but the too coarse resolution makes the analysis pretty weak. 

A: Having higher spatial resolutions for mobility data would be ideal to aid impactful information to public in relation to mobility patterns during a pandemic. Due to the privacy issue discussed in the Discussion section, this remains challenging. We stressed this issue as a methodological concern that should be addressed in future studies in terms of applications of big data such as social media data for public health. Please see that the last paragraph of the Discussion section, which was added to discussed this comment.  We added new sentences to note that higher spatial resolution would be helpful to understand neighborhood-level relationships between green space and mobility (lines 453-460).

Given this paper relies on free to access data and remote sensing, it would make sense to increase its robustness by enlarging considerably the study area. Why focusing only on a part of Maryland?

A: The FB Data for Good has gradually published open data for human mobility for more expended regions since March. However, those updated mobility data for some other regions in the US cover later periods during COVID-19 (e.g., ~ May) so there is a discrepancy for observation period with previous datasets. When we conducted our initial analysis for our study period, FB mobility data were available for some MCD groups in the East Coast area and Maryland was the only state for which mobility of MCDs was available for an entire state. We decided that using partial MCDs not representing their state, which likely has state orders for control measures, would also influence the robustness of our analysis and results. We added a brief explanation of this background to the Discussion (lines 441-446). Among the extended data of FB Data for Good, made available since our original analysis, California is the only state that has a close observation period to Maryland (e.g., Mar~) so we conducted analysis for California for this revision. We added new text to explain how future studies should be conducted using mobility data for other regions to confirm these results and address uncertainties. Please see lines 461-475.

It could be interesting to compare areas with different regulations regarding access to Outdoor settings. 

A: Thank you for this suggestion. We assume that different regulations and policies for access to green space or open spaces will modify the relationships between green space and mobility. This would lead to regional variability of such relationships. We responded to this comment in two ways. First, we added new analysis and results by region as suggested, by compare Maryland and California. Second, we added text discussing this issue as well. Please see lines 370-382.

The EVI is one possible index, having a better analysis regarding the amount of Green infrastructure would be required, not just pooling everything.

A: We agree that EVI is an index showing the density of healthy vegetation especially at a high-resolution such as a 250-m resolution used for the raw MODIS product, but does not fully capture the rich heterogeneity of green space.  Even though it can provide insights regarding amount of green infrastructure, it does not fully provide information such as type of green space, which is especially relevant for the behavior and/or human mobility in relation to use of green space. There exists no uniform data base that reflects all these characteristics of greenspace, and EVI is commonly used as a proxy. To provide more applicable information for which green space should be required for a given region, the index is recommended to be combined with other datasets such as land cover or locations of specific destinations of interest. We added this new context to Discussion to emphasize the values of using remote sensing data of vegetation in this type of research. Please see the last paragraph in the Discussion section that was newly added.  We also added information on local parks to give better context to green space, in addition to the original data sets we used including state parks.

At this stage, I found this study too preliminary for publications. 

A: We agree that results have key limitations and uncertainties, however due to the unprecedented nature of the pandemic, scientific evidence is urgently needed as decisions are being made now, and such decisions have enormous impact on public health. We substantially expanded the analysis to include a larger study region, by adding analysis for California. We also added text to highlight the reviewer’s point and note the various uncertainties and limitations of this work, the need for future work to confirm these findings, and specific topics for further study (lines 447-488).

The conclusions are only weakly supported by observations. Combining FB data with people enquiries regarding their behaviour during the restrictions could also strengthen considerably the analysis.

A: We agree that such data would be useful. Unfortunately, such data are not currently available combining the variables used for this study and other important information such as behavior changes. We consider this an area for our future research, in which we hope to develop individual-level data on why people visit, how often they visit, and how long they stay in green space. We consider researching such behavioral patterns as future research questions. We have added text to note the need for such data. Please see the last paragraph of the Discussion section stresses out how to improve methodologies and bring more practical applications.